# An Influence of Snow Covers on the Radar Interferometry Observations of Industrial Infrastructure: Norilsk Thermal Power Plant Case

Alexander Zakharov and Liudmila Zakharova *

Fryazino Branch of Kotelnikov Institute of Radio Engineering and Electronics, RAS, 141190 Fryazino, Russia
* Correspondence: ludmila@sunclass.ire.rssi.ru

**Abstract:** This manuscript presents the results of the study of snow covers' influence on the interferometric measurements of the stability of industrial infrastructure in the vicinity of Norilsk city, Russia. Fuel tanks of the Norilsk thermal power plant (TPP) were selected as an object of study due to a well-known accident when about 20,000 tons of diesel fuel spilled from one of the tanks. Sentinel-1 synthetic aperture radar data acquired over the territory of Norilsk TPP were used in the DInSAR study of the possible displacements of the tanks that could be the cause of the tank's damage. For twelve days, radar interferograms that were generated in the study covered the cold and warm seasons of 2018–2020, including the catastrophic event—the rupture of the tank with diesel fuel—in order to shed light on the possible impact of the area subsidence because of permafrost thaw under the tanks. As the tank walls and adjacent concrete base constituted the virtual dihedral corner reflector, the accumulation of snow on the surface near the tanks created a distorting effect on the results of monitoring the stability of the tank's location. Three models of snow layer within the dihedral proposed could help explain the deviations in the signal amplitude and phase in the case of snowfalls occurring between radar observations. We propose three ways to minimize the influence of snow on interferometric measurements. One of them, the selection of the radar data acquired in proper observation conditions, made it possible to assess the stability of the mutual location of the tanks. Among the most important processing and analysis results in the paper is a conclusion about the high stability of the fuel tank's location on the yearly time interval, including the troubleshooting tank.

**Keywords:** fuel tank; snow layer; freeze/thaw processes; synthetic aperture radar (SAR); differential SAR interferometry (DInSAR); Sentinel-1; Norilsk



## 1. Introduction

The oil spill because of the release of diesel fuel on 29 May 2020 from a reservoir owned by the Nornickel subsidiary Norilsk-Taimyr Energy Company at the Thermal Power Plant became an environmental disaster on a Federal scale in Russia. According to preliminary estimates, 6000 tons leaked into the ground, and 15,000 tons flew down into the nearby Ambarnaya and Daldykan rivers [1]. According to the Federal Service for Supervision of Natural Resources, the level of harmful substances in the Ambarnaya River on 3 June 2020 exceeded the maximal permissible concentration tens of thousands of times. The further propagation of polluted waters to the North created a threat of pollution in Lake Pyasino, the Kara Sea, and the Arctic Ocean, successively. Only one year later, in August 2021, the Ministry of Emergency Situations announced the elimination of the consequences of the fuel spill. Permafrost thaw was called by Nornickel authorities as the main and the only reason for the accident: unusually warm weather could soften the permafrost and affect the state and stability of the concrete base and piles under the tank, that triggered the displacement of the tank and led to the subsequent accident. Greenpeace Russia put out a statement refuting the assertion that permafrost caused the Norilsk oil spill: "The fuel tank

failed due to ulcerative corrosion, which caused holes to form at the bottom of the tank and led to the tank's eventual collapse". SAR interferometry is the technique that could reveal the fact of the subsidence of the scattering covers before the accident, which could cause damage to the tank.

Differential SAR interferometry (DInSAR) as a tool for detecting and registering surface deformations from space has evolved since the end of the 1980s [2]. Nowadays, advanced methods exist, such as Permanent Scatterers [3] or Persistent Scatterers [4], and their numerous modifications are widely used for monitoring displacements of the Earth's surface. Modern SAR instruments with medium and high spatial resolution allow the exploration of smaller and smaller scattering objects; thus, now we can explore not only the buildings as a whole but also in detail and create 3D models of their relative displacements [5]. As man-made structures need close attention for accident-free operation and public safety, they quickly became the subject of interferometric studies carried out by numerous researchers all over the world; this includes the detection of surface subsidence in mining areas [6–8], as well as urban area displacements caused by subway construction and operation [9–11], and bridges, viaducts, and dams stability assessing [12–16]. Landslides, permafrost thaw, and the consolidation of sediments under heavy constructions lead to subsidence that affects urban and infrastructure units. Interferometry provides a powerful instrument for monitoring various objects at risk of displacement: railways [17,18], city blocks and individual buildings [19,20], highways [21], and airport runways [22,23].

This study aims the demonstration of restrictions in the application of the DInSAR technique for monitoring industrial infrastructure in the Arctic region. We show that the formal application of DInSAR techniques for the monitoring of the structures producing double bounce scattering may give incorrect results because of the corrupting impact of atmospheric precipitation (most particularly snowfalls), and we discuss ways to minimize the influence of corrupting factors. The paper is organized as follows. Section 2 describes the study area. Section 3 describes the dominant scattering mechanism of industrial infrastructure elements such as fuel tanks located in the study area. Section 4 presents three possible models of the snow layer covering the surface nearby the tanks. Section 5 describes processing methods. Section 6 reports the results of the SAR data processing and analysis of results, and Section 7 provides the conclusions.

## 2. Study Area

The Norilsk city is located south of the Taymyr Peninsula, Russia, 300 km north of the Arctic Circle and 2400 km from the North Pole. The climate of the Norilsk region is of a subarctic type with very long, extremely cold winters and very short, mild summers. Negative air temperatures are observed 240 days a year. Stable snow cover forms in the first half of October and disappears until the second decade of May. The average annual air temperature here is $-9.6\,°C$, with an annual variation in absolute temperatures of $85\,°C$. The dominant winds of the southern quarter in winter are the cause of the transfer of large masses of snow, the formation of deep snowbanks, and sastrugi on the snow surface. The region is characterized by continuous permafrost. Tundra gley soils, marshes, and alluvial soils are the most typical here.

The TPP area ($69.326°N$, $87.935°E$) is adjacent to the metallurgical plant named after B.I. Kolesnikov from the southwest, constituting a common industrial zone on the edge of the Nadezhda Plateau. Google Earth images of Norilsk city's surroundings with TPP in the middle (marked with a yellow pin) are presented in Figure 1. The inset top right shows an enlarged fragment of the image with a line of fuel tanks oriented diagonally. The tanks are numbered from bottom to top, from tank 2 to tank 5. A circular footprint of the dismantled tank 1 below can be seen also. The yellow arrow indicates the Sentinel–1 satellite flight direction in SAR data acquisitions, which are available from ESA archives, and the broad blue arrows indicate the onboard synthetic aperture radar (SAR) observation direction. In such observation geometry, the western walls of the fuel tanks are shadowed, as well as the

adjacent parts of the underlying concrete base. Consequently, the location of the oil leak from troubleshooting tank 5 (indicated by the white arrow) is hidden from the radar.

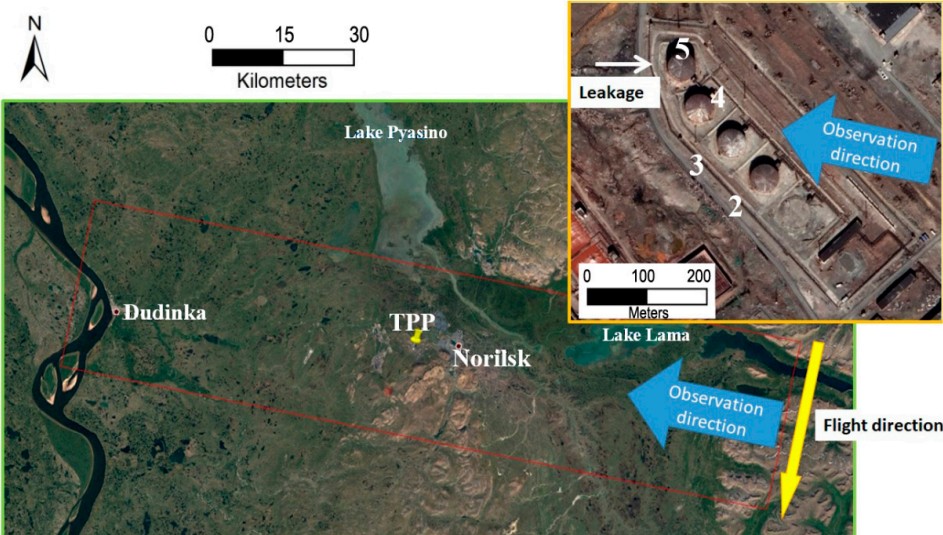

**Figure 1.** Google Earth image of TPP area. White numbers 2–5 on the inset enumerate fuel tanks.

The fuel tanks in Figure 2 (Available online: https://polarjournal.ch/en/2020/06/05/severe-environmental-disaster-in-norilsk/, accessed on 13 November 2022) are cylindrical structures with cone-shaped roofs with a radius of $r = 20$ m; they are spaced at a 70 m distance. The territory is surrounded by a concrete fence, and concrete partitions separate one tank from another. The tank's height $h$ is 20 m (tanks 2, 3, 4) and 30 m (tank 5). The roof slope is below $20°$. In the image taken after the catastrophic event, the roof of the troubleshooting tank 5 (the left one in Figure 2) is sucked in due to the under pressure created by the fuel leak.

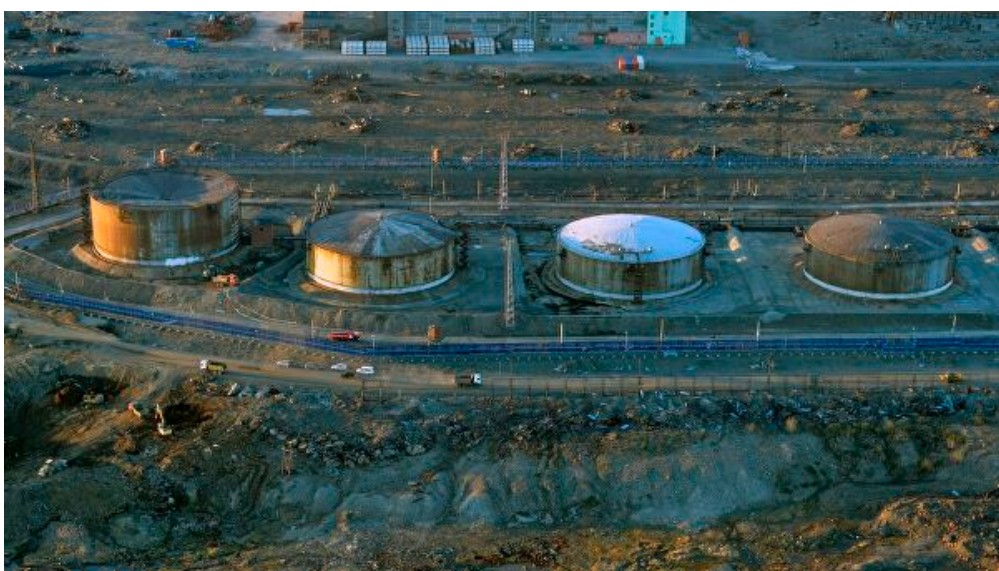

**Figure 2.** Picture of TPP fuel tanks. View from south-west.

## 3. Tank Scattering Model

The radar images of such industrial objects differ from optical ones significantly. The main type of radio wave interaction with the tank wall and roof, which are smooth at a wavelength scale, is a mirror-like scattering. The roof is unseen in radar images, as it scatters the signal S away from the radar (see Figure 3). The cylindrical body of the tank on

the concrete base is similar to a classical reference target, along with the top-hat reflector. A narrow vertical strip of the cylinder surface with a width of $w$ and height of $h$ is the only part of the cylinder responsible for the signal scattering in the radar direction. A well-known expression for the radar cross section (RCS) of the cylinder in the case of signal wavelength $\lambda$ is:

$$\sigma_c = \frac{2\pi r h^2}{\lambda}. \tag{1}$$

The strip constitutes a sort of dihedral corner reflector AOB with a horizontal surface near the tank. The vertical plate AO of the dihedral is marked with yellow in Figure 3; its width is $w = 0.74$ m, according to the next formula derived in a supposition of the $\lambda/8$ variation in the signal two-way path length along the strip width:

$$w = \sqrt{0.5 r \lambda}. \tag{2}$$

Signal D in Figure 3 experiences double bounce scattering on the dihedral plates and propagates in a backward direction to the radar. The size of the virtual horizontal plate of the dihedral (plate OB) is determined by the size of the adjacent vertical plate on the cylinder surface (plate OA) and observation geometry. The specificity of the signal scattering from the dihedral corner is that all the signals interacting with its plates have the same two-way propagation distance. For that reason, on the radar image, we can see the tank as the only bright point object at the location of the dihedral vertex O instead of the extended feature, despite a high range resolution (2.3 m) of Sentinel-1 SAR compared to the tanks walls height (20 and 30 m). The RCS of this virtual dihedral corner with ideal scattering properties of its plates is:

$$\sigma = \frac{16\pi S_p^2}{\lambda^2} \sin^2\theta, \tag{3}$$

where $S_p$ is the square of the dihedral plate and $\theta$—incidence angle ($\theta \leq 45°$).

The dihedral mainlobe width in the vertical direction at $-3$ dB level is ~30° [24]. If angle $\theta = 45°$, then the theoretical RCS of tank 5 is 65.4 dBm², and it is 61.9 dBm² for the other tanks.

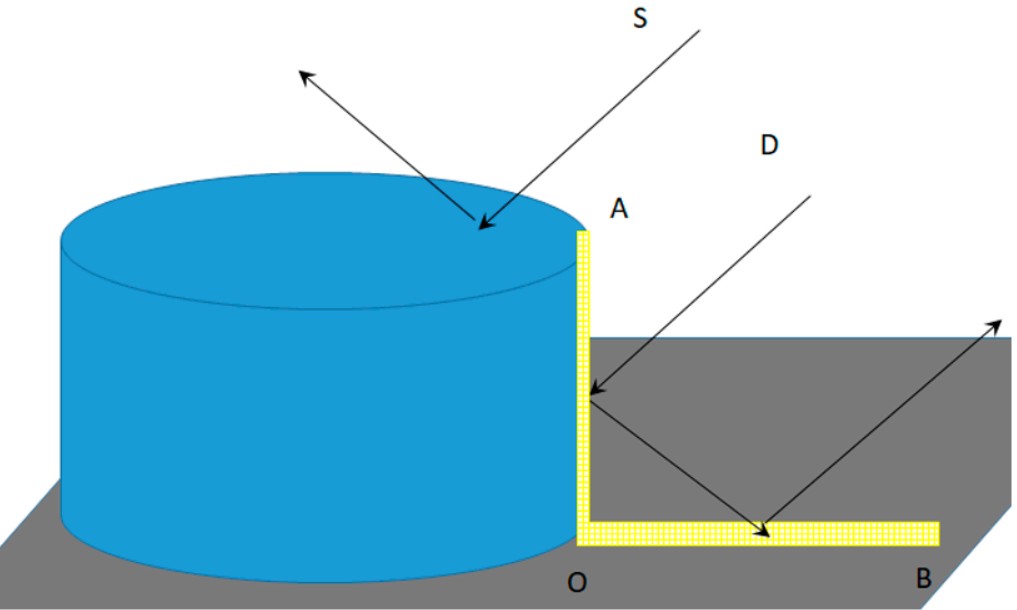

**Figure 3.** Interaction of radio waves with the tank surface: S—single scattering, D—double bounce scattering. AOB—dihedral corner reflector.

## 4. Snow Layer Models

The state of the scattering surface within the horizontal plate OB of the dihedral (see Figure 3) is essential for the most correct interpretation of interferometric phase measurements. For example, the scattering properties of wet covers are different from dry ones; thus, the level of the backscattered signal may vary as well as the location of the dihedral vertex O. In winter, the main source of alteration for the propagating signal properties is the growth in snow depth between the SAR observations.

### 4.1. Uniform Snow Layer on the Dihedral Plate

The influence of the snow layer with uniform thickness $s$ on the radio signal propagation is commented on in Figure 4. Here, $\Delta R_s$ is a signal path in snow-free conditions, $\Delta R_a + \Delta R_e$ is a signal path in the presence of the snow layer, and $\theta_i$ is a signal incidence angle [25]. Guneriussen et al. gives the following simple relation between the snow depth $s$, two-way path increment $\Delta l_s$, and phase difference $\Delta\varphi_s$ on the interferogram:

$$\Delta l_s = \frac{\lambda \Delta\varphi_s}{4\pi} = s\left(\cos\theta_i - \sqrt{\varepsilon_s - \sin^2\theta_i}\right). \tag{4}$$

where $\varepsilon_s$ is the permittivity of snow. Dry snow permittivity is related to the snow density $\rho_d$ as [26]:

$$\varepsilon_s = 1 + 1.6\rho_d + 1.86\rho_d^3. \tag{5}$$

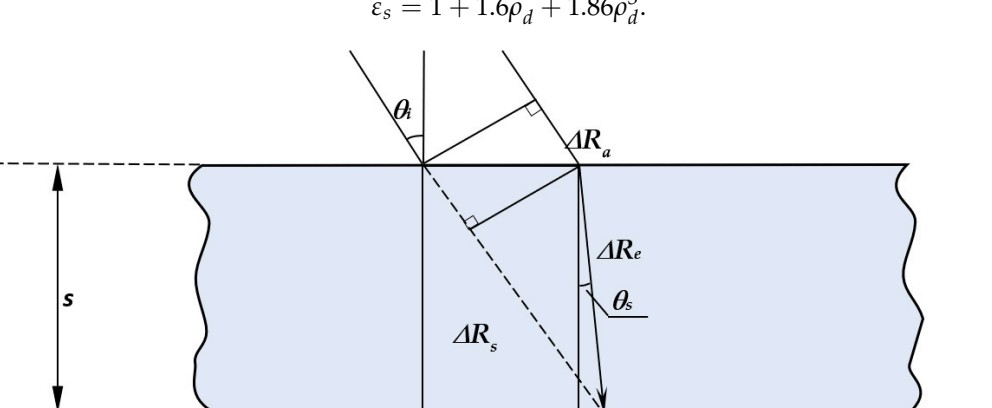

**Figure 4.** Signal propagation through the snow layer on the flat surface.

In the case of a 3 cm increment in the fresh snow layer with $\rho_d = 0.3$ g/cm$^3$ between SAR data takes, the signal one-way path increment would be 1 cm for the incidence angle $\theta_I = 41°$. The corresponding phase increment would be ~2 rad in C-band. Such a phase alteration may be misinterpreted as a 1 cm subsidence in the adopted scheme of interferometric processing. The unambiguous reconstruction of snow thickness $s$ or potential slant range subsidence $\Delta l_s$ from $\Delta\varphi_s$ in (4) is possible, provided that the $2\pi$ ambiguity of the phase measurements is resolved.

### 4.2. Wedge-Shaped Snow Layer Inside Dihedral

Dry snow is transparent at microwave frequencies, but an uneven snow layer upon the OB plate may decrease the level of the signal backscatter because the plates of the dihedral can act as non-orthogonal ones. As a result, the alteration of the dihedral pattern would cause the deflection of the backscattered signal direction. The case of the wedge-shaped snow layer on the dihedral plate with a wedge angle $\alpha$ is presented in Figure 5. Let the incidence angle of the incoming signal be $\theta_1$ and for the outgoing signal be $\theta_5$, and the relation between the angles would be as follows:

$$\theta_5 = \theta_4 - \alpha = \arcsin\left(n\sin\left(\arcsin\left(\frac{\sin(\theta_1 - \alpha)}{n}\right) + 2\alpha\right)\right) - \alpha. \tag{6}$$

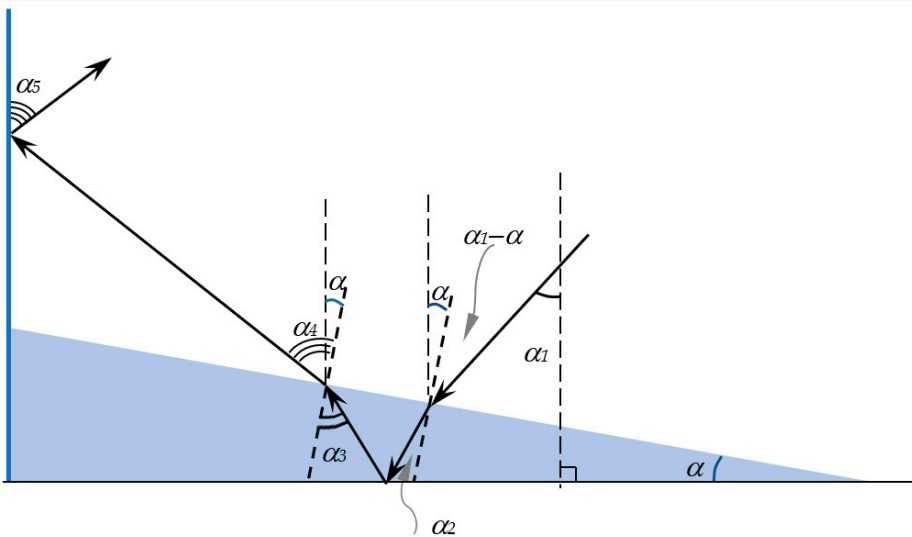

**Figure 5.** Signal propagation through the wedge-shaped snow layer within the dihedral.

If the refractivity coefficient of the snow layer is $n = 1.25$ and $\theta_1 = \frac{\pi}{4}$, then for the angle $\alpha = 1°$ $\theta_5 - \theta_1 = 0.9°$, and for $\alpha = 10°$ $\theta_5 - \theta_1 = 11.2°$. We can see that even for small wedge angles, the backscattered signal misses the radar. Therefore, in the presence of the wedge-shaped snow layer, an ideal dihedral acts as a dihedral with non-orthogonal plates. As a result, the dihedral pattern is split into two narrow-shaped patterns of the dihedral plates, with maximums of the main lobes declining from the SAR direction [24]. In more realistic conditions, the oscillation in the orientation of numerous facets constituting the snow surface would smear the dihedral pattern in the angular direction and decrease the backscatter level.

*4.3. Heterogeneous Snow Layer Inside Dihedral*

Among the realistic scenarios of the snow coverage on the horizontal plate OB of the dihedral (Figure 3) is the presence of snow-free patches, where the snow may be blown away by strong winds. Let the bistatic normalized RCS of the concrete surface within the boundaries of the OB plate be equal to $\sigma_b$, and $S_0$ represent the snow-free square of the OB plate with square $S$. Taking in mind that there is no attenuation of C-band signals traveling in the snow layer of an order of meter-scale thickness, we may write out the phase of the entire signal scattered from the plate OB as:

$$\Delta\varphi_d = Arg\left(\sqrt{\sigma_b}\frac{S_0}{S}e^{j\varphi_0} + \sqrt{\sigma_b}\frac{S-S_0}{S}e^{j(\varphi_s+\varphi_0)}\right). \tag{7}$$

The first term in the brackets of (7) is a signal of the snow-free surface, and the second one is a signal of the surface under the snow layer. Phase $\varphi_s$ here describes the influence of the snow layer on the phase of the propagating signal, and phase $\varphi_0$ is a phase of backscatter from the plate OB. The latter one can further be considered as zero. In the plot below, the dependence of the phase $\Delta\varphi_d$, the entire signal echo from the plate with various percentage contributions of the snow-free surface as a function of the phase introduced by snow layer $\varphi_s$ is shown. In the case of the entire coverage of the plate with snow (0% input of snow-free surface), the phase of the signal is equal to the phase of the signal propagating through the uniform snow layer. Consequently, it is possible to evaluate snow depth correctly according to (4), providing that the phase ambiguity of the phase measurements is resolved correctly. With the increase in the square of the snow-free area, the direct dependence breaks (see Figure 6). In the case of the 50% square of snow-free area, the phase $\Delta\varphi_d$ varies within the $\pm 1.2$ radian only, and in the case of 75% within the $\pm 0.4$ radian. In such a case, a correct procedure of $2\pi$ phase unwrapping is impossible, as

well as the restoration of the full phase introduced by the snow layer and evaluation of the displacements of the fuel tanks scattering constructions.

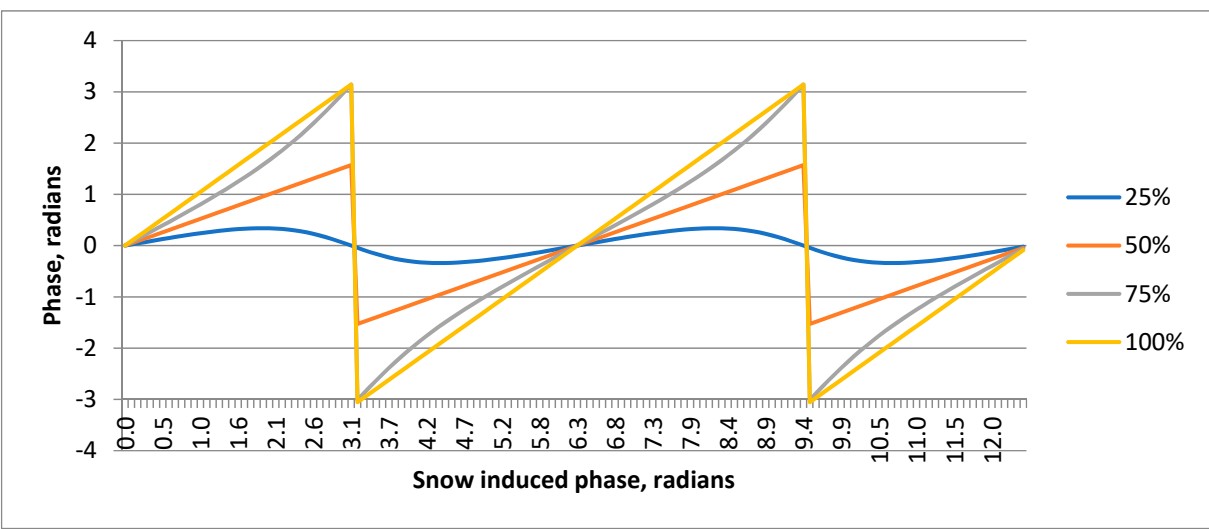

**Figure 6.** Phase of the signal scattered by the plate OB with respect to the snow induced phase for different percentages of snow cover.

The three above-mentioned models of the influence of the snow layer can help to explain the mechanism of the influence of snow on the deviations of the amplitude and phase of the signals scattered from the tanks.

## 5. Methods

The typical Sentinel-1 SAR image of the Kolesnikov metallurgical plant territory and respective interferogram for the period 3 May 2020–6 June 2020, covering the catastrophic event occurring on 29 May 2020, is presented in Figure 7. Thanks to high RCS, the tanks are visible as bright point objects; their location is marked with a red ellipse. In our earlier study [27], we looked for small-scale deformations or displacements on the interferograms with a short temporal baseline. This 24-day interferogram contains discernible variations of the interferometric phase difference across the plant territory, which is mainly caused by soil thaw processes in the spring and, probably, atmospheric irregularities. We did not find any manifestation of the displacements of the tanks constructed within the red ellipse on this interferogram. According to our estimations, the mutual tank position was fixed within the 2–3 mm tube [27].

Probable earlier displacements of the reservoir constructions, if they were, could lead to an increase in the tension within the tank walls and finally cause damage to tank 5, so at the first stage, it was decided to select a longer monitoring interval—from July 2019 until August 2020. Generally, the accepted approach in the monitoring of the steady small-scale surface movements in the case of high temporal decorrelation is an application of the persistent scatterers (PS) technique or similar ones, as mentioned in the Introduction section [4]. The main feature of the PS technique is the identification of stable scatterers via the statistical analysis of the large time series of the images. The criterion for the selection of candidates for persistent scatterers is the stability of the signal backscatter. Our analysis of the tanks' backscatter during a one-year interval showed that it did not satisfy the stability criterion from [28–31]. The ratio of the signal standard deviation to the mean for tanks 2−5 was 0.6, 0.93, 0.81, and 0.98, respectively. Among the possible reasons for such high instability is the influence of atmospheric precipitation as well as freeze/thaw processes on the underlying covers. Inevitable abrupt alterations of the location of the phase center of the backscatter due to alterations in the soils' radio physical properties differ from slow and monotonous changes assumed in the PS technique. Therefore, the PS technique cannot be considered a reliable tool for the detection and evaluation of TPP fuel tank

instabilities. For that reason, a classical differential interferometry approach was applied here to measure movement-induced phases, and the measurements were performed at the reservoir locations that had a maximal level of the backscatter on the amplitude images.

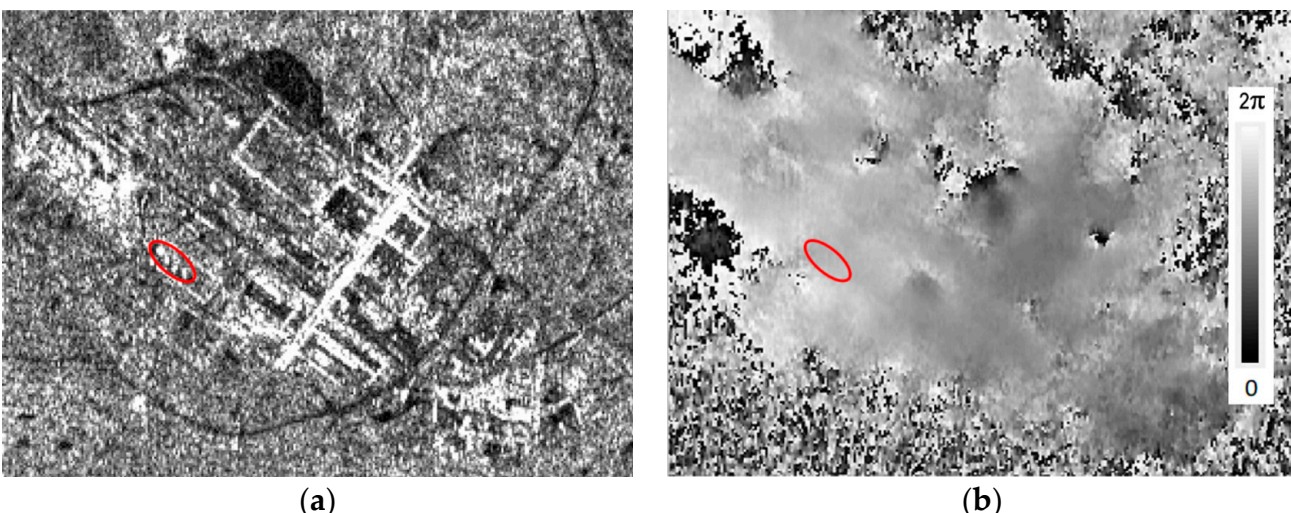

(**a**)  (**b**)

**Figure 7.** Sentinel-1 SAR images of the Kolesnikov metallurgical plant and thermal power plant territory with encircled fuel tank's location: (**a**) Amplitude image; (**b**) Interferogram covering the catastrophic event.

SAR interferometry techniques exploit the phase information of the echo signals registered by the radar system at two close points in space, provided that the mutual coherence of signals is satisfied. The signals phase difference variations in one-pass scheme are dependent on the variations of signal range differences as a function of underlying surface topography variations, and in the two-pass scheme—on the variations in the signal propagation conditions between SAR observations as well [32]. Interferogram represents a map of the phase differences of the signals $U_1$ and $U_2$ scattered from the same target and acquired at two acquisition points. The mathematical expression for a complex interferogram is as follows:

$$U_1 U_2^* = u_1 u_2 e^{j(\varphi_1 - \varphi_2)} = u_1 u_2 e^{\frac{-4j\pi\Delta r}{\lambda}}, \tag{8}$$

where $u_1$ and $u_2$ are the amplitudes of signals, $\Delta r$ is the difference in slant ranges from observation points to the surface object.

The quality of the interferometric phase measurements is dependent on the coherence of the backscattered signals. The complex correlation coefficient or coherence $\gamma$ between the two complex SAR images $U_1$ and $U_2$ is defined as:

$$\gamma = \frac{E\{U_1 \cdot U_2^*\}}{\sqrt{E\{|U_1|^2\} E\{|U_2|^2\}}}, \tag{9}$$

where $E\{\}$ denotes the expectation value operator over the array of the nearby independent image samples used in the estimation. An analytical expression for the phase as a function of the coherence, according to the Cramer-Rao lower bound [33], is:

$$\sigma^2(\Delta\varphi) = \frac{1}{2N_L} \cdot \frac{1-\gamma^2}{\gamma^2}, \tag{10}$$

where $N_L$ is the number of independent pixels on the interferogram used to derive the phase and is usually referred to as the number of looks. As can be seen, the larger the coherence, the lower the phase dispersion.

The phase difference $\Delta\varphi$ includes (at least) the topographic phase $\Delta\varphi_{topo}$, the phase $\Delta\varphi_d$ generated by small-scale displacements of the underlying surface (surface dynamics), atmospheric phase fluctuations $\Delta\varphi_a$ caused by variations in the signal path length of the atmosphere, extra phase component $\Delta\varphi_s$ in the snow layer, as well as random phase fluctuations caused by thermal noise $\Delta\varphi_n$, spatial $\Delta\varphi_{spat}$ and temporal decorrelation $\Delta\varphi_{temp}$, and, finally, the unknown initial phase. The typical decomposition of the interferometric phase is as follows:

$$\Delta\varphi = (\Delta\varphi_{topo} + \Delta\varphi_d) + (\Delta\varphi_a + \Delta\varphi_s + \Delta\varphi_0) + (\Delta\varphi_{spat} + \Delta\varphi_{temp} + \Delta\varphi_n). \tag{11}$$

First, brackets in (11) contain phase components describing the information of interest, i.e., surface topography and small-scale dynamics. Thermal noise and the noise of spatial and temporal decorrelation in the third brackets affect the interpixel accuracy of the entire phase measurements. These corrupting noise-like components may be suppressed via interferogram spatial filtering. Atmospheric phase variations caused by spatial irregularities in the refraction index as well as a snow component in the second brackets are usually slow varying spatial patterns; they may be estimated on the nearby knowingly stable test targets and subtracted. In this procedure, the unknown initial phase $\Delta\varphi_0$ was also removed.

In this study, a series of Sentinel-1B C-band SAR images (wavelength $\lambda$ = 5.6 cm) covering the two years time interval of SAR observations were used. The images were acquired from the descending orbit in the right-side looking acquisition geometry with ~41° signal incidence angle. Interferometric SAR data processing was performed in the ENVI SARscape processing environment. The topographic phase from the first brackets of (11) was estimated using SRTM-X data with 30 m surface spacing and was subtracted from the interferometric phase [32]. In order to suppress the noises of various natures, the multi-looking procedure was applied with subsequent interferogram filtering by the Goldstein filter [34] with an effective spatial window size of 5 × 5 pixels. The total phase of a group of corrupting components from (11)—the atmosphere, snow, and signal initial phase—was estimated at the location of the most stable scatterer nearby (to be discussed later) and subtracted.

Free of the topographic phase and most other corrupting components from (11) the remaining surface dynamics phase $\Delta\varphi_d$ can be converted to the line-of-sight (LOS) displacements of the scattering surface $\Delta r_d$ as:

$$\Delta r_d = -\frac{\lambda}{4\pi}\Delta\varphi_d \tag{12}$$

## 6. Results and Analysis

Meteorological conditions in the observation area are of high importance for the interpretation of interferometric measurements. Figure 8 shows a plot of the average air temperature and snow depth recorded by the Norilsk weather station from August 2019 until August 2020 (weather data is available online www.rp5.ru, accessed on 4 January 2023). The red line is the air temperature in degrees, and the blue one is the snow layer thickness in centimetres. At the beginning of September 2019, the air temperature crossed the zero level. The short-term cold snap and first snowfall occurred on 13 September 2019. Permanently negative temperatures and the formation and accumulation of snow covers were established since mid-October. The snowmelt period began in mid-April and ended in May. Such strong variations in the state of the scattering cover because of freeze/thaw are known to be a source of measurement errors in SAR interferometry.

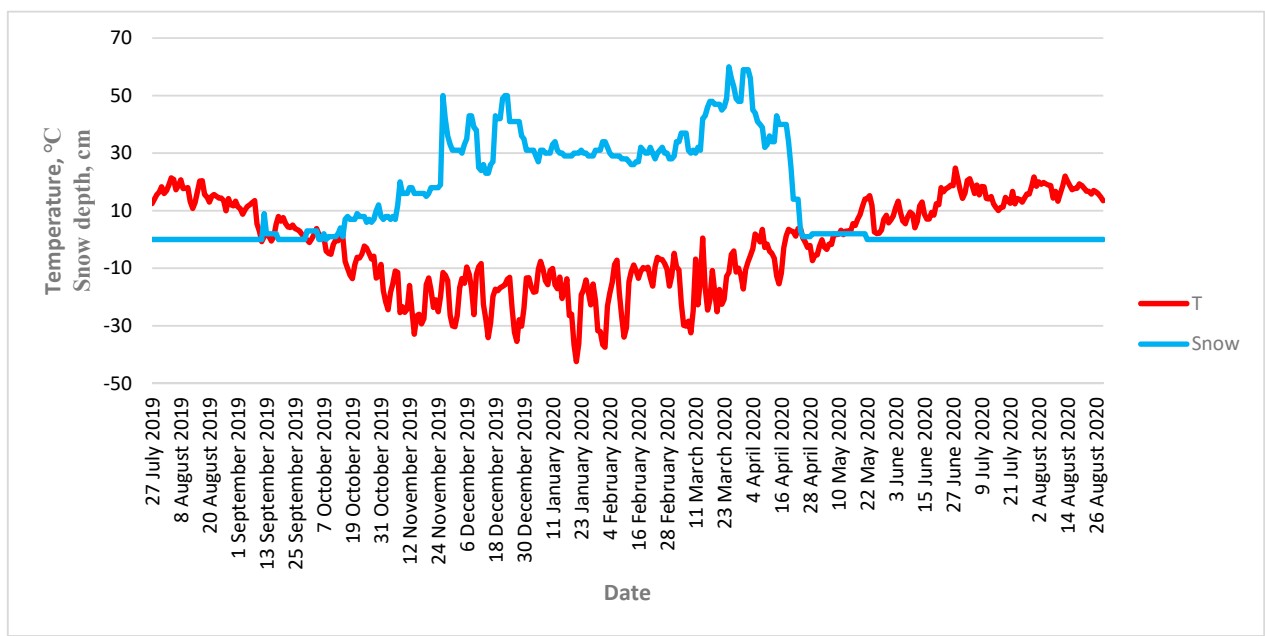

**Figure 8.** Air temperature and snow depth in 2019–2020 at Norilsk weather station.

In Figure 9, the variation in RCS for tanks 2–5 during the same time interval mentioned in Figure 8 is shown. Observation dates are given here along the horizontal axis. In general, the RCS of all the tanks is ~30–35 dBm$^2$ below the theoretical level. This can be explained by the imperfection and non-orthogonality of the walls of the tanks and the the concrete base constituting the dihedral corner reflector, as well as low levels of the bistatic RCS of concrete base within rectangle OB. The first noticeable decrease in the backscatter level was observed on 13 September 2019, during the first short-term cooling and snowfall. Later on, steady snow layer accumulation since 19 October 2019 did not cause prominent changes in the backscatter of all the tanks, although their RCS fluctuations were too large to be explained by the alteration of the thickness of the snow layer on the plate OB, especially for tank 3 in February, when the tank RCS decreased by ~10 dB. The most probable reasons for backscatter fluctuations are the non-orthogonality of the plates caused by wedge-shaped snow and the phase fluctuations because of fluctuations in the snow layer thickness along the plate OB. Then, the level of the tank's backscatter increased in April 2020, at in the beginning of the snowmelt period. The abrupt decrease in the RCS of tanks 4 and 5 at the end of July 2020 was explained by dismantling these tanks. The amplitude dispersion index [4] of tank 2–5 signals from Figure 9 was too high from a formal point of view to consider the tanks as candidates for persistent scatterers and to apply the PS techniques to monitor the tanks' displacements in wintertime.

To study the long-term dynamics of the TPP fuel tanks during the period August 2019–August 2020, covering potentially small-scale steady subsidence effects, we computed coherence maps and differential interferograms with 12day intervals between SAR observations. The topographic phase was estimated using GMTED2010 DEM (USGS & NGA, USA) and subtracted. To correct for the unknown initial phase $\varphi_0$ we chose a stable reference object and natural corner reflector formed by the right angle of the administrative building and the surface of the ground (the corner is located at the distance 1050 m in the east-south-east direction from the tanks). The phase of the stable reference target was subtracted from the phases on the interferograms, and the remaining phase difference measurements were accumulated over time starting from July 2019.

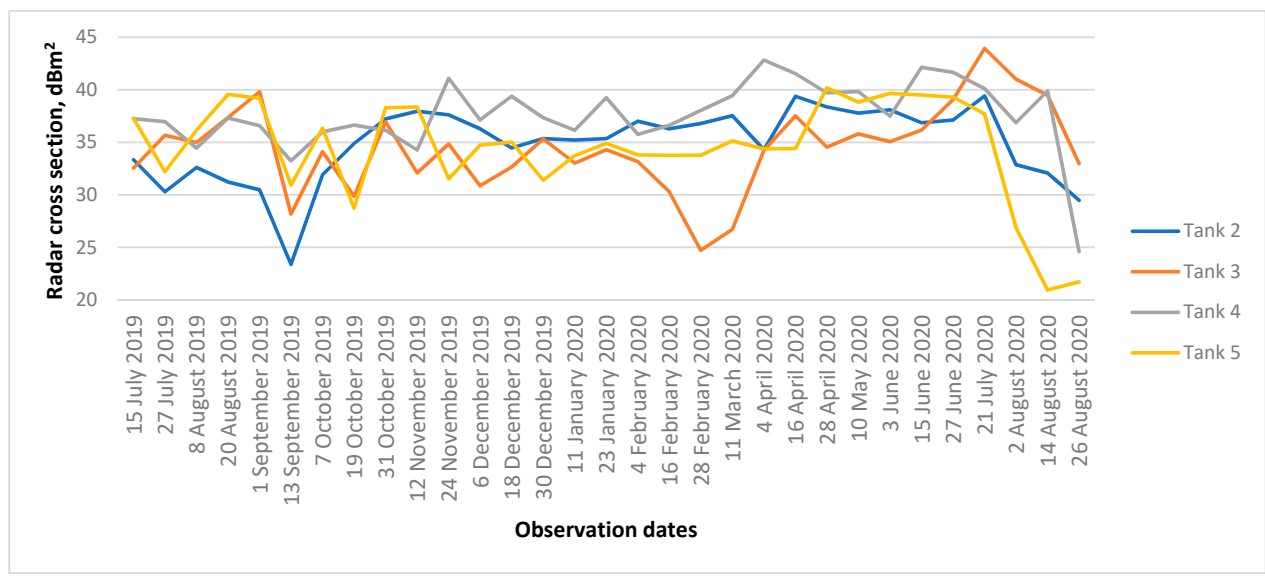

**Figure 9.** RCS of tanks, dBm², in Sentinel-1 data for 2019–2020.

The plot of the twelve-day coherences is presented in Figure 10. As one can see, the coherence is much more sensitive to alterations in the radio's physical properties than the signal intensity. First, the prominent decrease in the coherence occurred on 13 September. The coherence of all the tanks remained low until the start of the period of stable negative daily temperatures. Snowmelt in April also produced an explainable drop in coherence. Finally, coherence in the area of tanks 4 and 5 dropped sharply because of their dismantling at the end of summer.

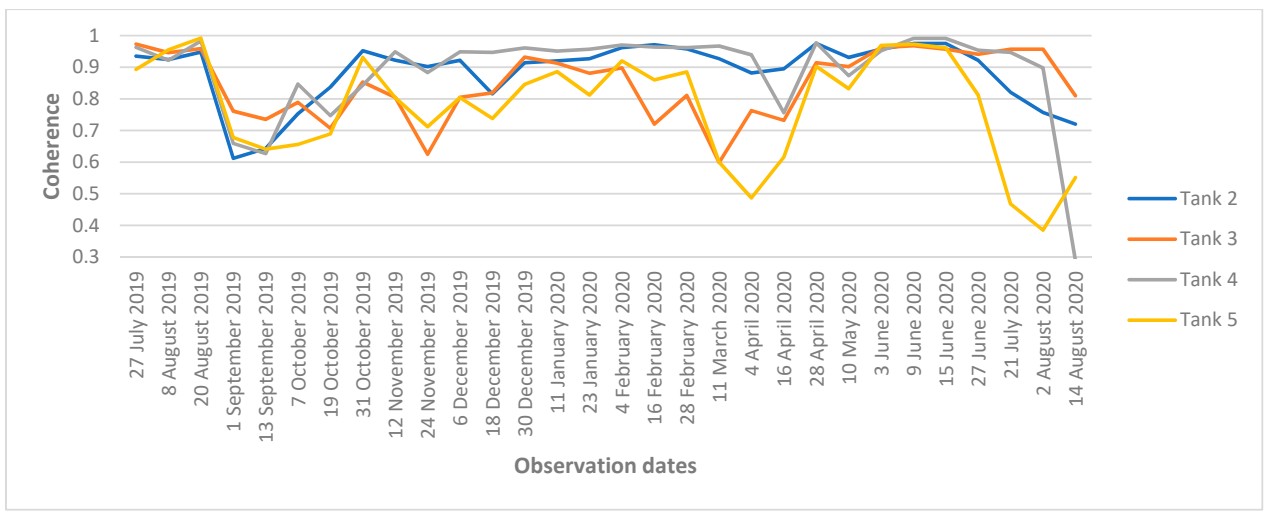

**Figure 10.** Interferometric coherence (no units) of tanks backscatter in 2019–2020.

The plot of twelve-day interferometric phase differences is presented in Figure 11. The synchronicity in phase alterations is violated in the case of precipitation and freeze/thaw processes. Cold weather on 13 September, snowfalls in November–December, and snowmelt in April are also the cause for the divergence of the tank's phases.

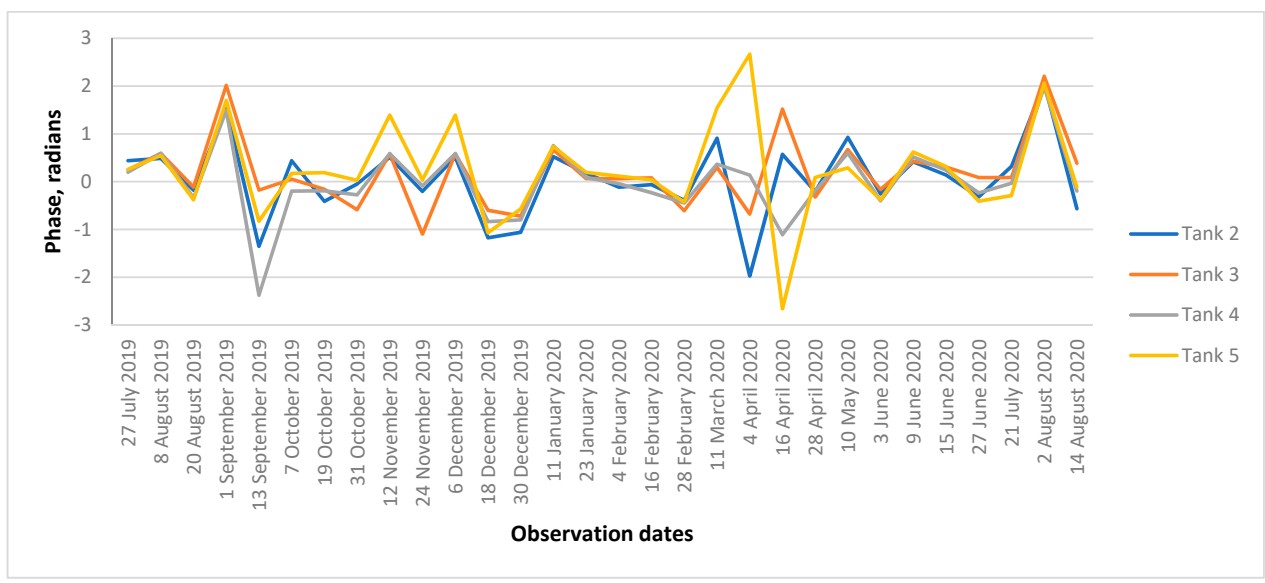

**Figure 11.** Interferometric phase for 12 day intervals in 2019–2020.

In order to reveal the possible long-term and small-scale displacements of the tanks, the phases from Figure 11 were accumulated sequentially, and the plot of the accumulated phases since 27 July 2019 is presented in Figure 12. As can be seen here, the phase of all the tanks was close to zero during August 2019. The first sharp phase deviation from zero occurred on 13 September. In the adopted scheme of interferometric processing, such a phase increment could be misinterpreted as the subsidence of the scattering surface if ignoring the fact of the regular phase deviation of the signal propagating through the snow layer. Further, throughout the winter, an increase in tank 5's phase can be seen. It grows (represented by the yellow line) until the start of the snow melting period in April, then it drops to almost the zero level after the end of the snow melting period in May. Such phase variations could be interpreted as monotonous subsidence of tank 5 during the winter with a return to its original position in spring, if not taking into account the influence of snow. As for another tank, the phase of tank 4 decreases until the $-2$ radian, as if there was an uplift at the 0.9 cm level. The phase of tank 3 grows till $+3$ rad (a 1.3 cm subsidence?), and it oscillates around the zero value for tank 2.

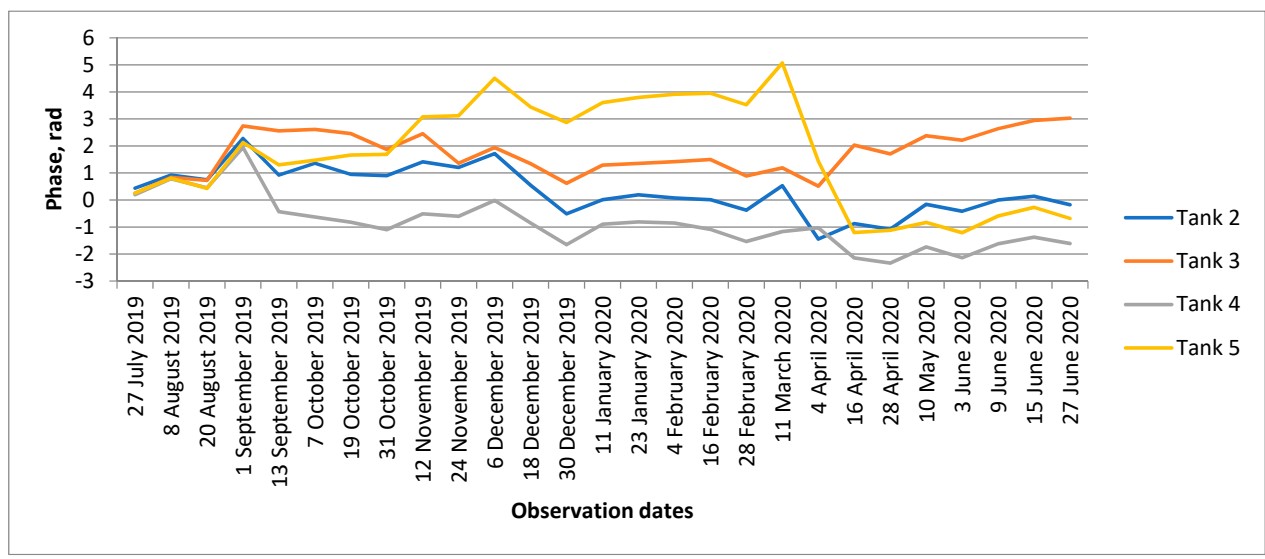

**Figure 12.** Accumulation of the phases for tanks 2–5 in 2019–2020.

Despite the fact that there is a simple relationship between the increment in the two-way signal path length (or phase) and the increment in snow layer thickness (4), usually it is impossible to obtain sufficiently accurate phase corrections using snow depth data from nearby weather stations only. Although the weather station of Norilsk city reported a 50 cm snow depth by the end of winter 2019–2020, the accumulated phase of troubleshooting tank 5 reached ~5 rad only. Despite this, there is some correlation between the plot of the phase of tank 5 in Figure 12 and the snow depth in Figure 8. Snow accumulation since 10 October was noticeable as a phase rise (yellow line in Figure 12). An abrupt drop of the tank 5 phase until zero values in April was in agreement with the snow melting process at that time. As for the other tanks, there was no reasonable correlation between their phases and snow depth variations.

In Figure 13, there is a Google Earth sample optical image of the tanks acquired at the beginning of spring. Yellow strips mark the location of the horizontal plates of dihedrals that participate in the process of the double-bounce scattering of the SAR signal toward the radar. The specific feature of this early spring image is the presence of snow covers on almost all the territories of the TPP and its surroundings. Prevailing winds of the southern quarter in winter here are a common cause of the transport of large snow masses in Norilsk; the winds blow out snow covers in open areas and sweep deep snowbanks near the barriers. Red arrows on this sample image point out darker areas where the snow has been blown away. Eastern (right) parts of the yellow rectangles are covered with snow. The retention and accumulation of snow on these distal parts of the plates persist through the winter thanks to concrete division fences around the tanks working as snow fences.

As it follows from the plots in Figure 7, in the presence of the snow-free area within the image pixel, the relation between the interferometric phase and snow-induced phase becomes non-linear. Additionally, the larger the snow-free area within the horizontal plate of the dihedral, the less the phase increment of the scattered signal for the same snow-induced phase. If the snow-free area occupies more than 50% of the square of the plate, the phase of the scattered signal oscillates within some narrow corridor. That is probably why the phases of tanks 2–4 oscillate around the zero level. The specificity of troubleshooting tank 5 is that it is 10 m higher than other tanks. For that reason, its virtual horizontal plate is longer, and the influence of the same snow-free area in winter 2019–2020 is lower. As we can see, due to many possible reasons—the wind transport of snow in open areas, local snowfall heterogeneity, as well as snow clearing in the survey area—the dynamics of snow cover accumulation in winter and its influence on the interferometric phase measurements can be judged only qualitatively.

One more example of the impact of snow on the interferometric measurements of the TPP tanks in the previous winter of 2018–2019 is presented in Figure 14. Four plots describe the accumulation of the same tank phases from 14 June 2018 to 8 August 2019. All the phases coincide with each other until 24 October 2018: the beginning of the interval of permanently negative daily temperatures and accumulation of snow covers. Since then, the plots diverged and oscillated around the zero value. Though the snow depth reached 80 cm by the end of winter 2018–2019 and accumulated in the interferometric phases of the tanks, it did not exceed two radians. Such a moderate accumulation of the phase may also be explained by the wind transport of snow and heterogeneity of the snow layer on the concrete base near the tanks.

The corrupting impact of atmospheric precipitation on the phase of reference targets has been mentioned in many studies. The influence of the water inside the corner reflector was referred to in [35], and the loss of the measurements after the snowfalls was described in [36]. One of the ways to avoid corrupting influences of snow cover accumulation and soil moisturizing is to work with the images acquired during snow-free seasons of the year and preferably in dry weather conditions. So far, to check the non-zero phases of tanks 3 and 4 from Figure 12, that accumulated by midsummer 2020, we processed 18 interferometric pairs with temporal baselines for longer than half a year. The images constituting the interferometric pairs were acquired in the warm seasons of the year, and the pairs spanned

the wintertime period. From the experience presented in a number of works, it followed that the proper selection of reliable phase measurements was desirable. In our study, for better accuracy of the phase measurements, we excluded the cases with coherences below 0.8. As a result, we excluded the pairs containing summertime observations made on rainy days or the dry days that followed 5 days intervals with 4 mm or more total precipitation. In Figure 15, the phases of all the tanks from 13 interferograms with long temporal baselines (from 6 to 12 months) from July 2019 to August 2020 are presented.

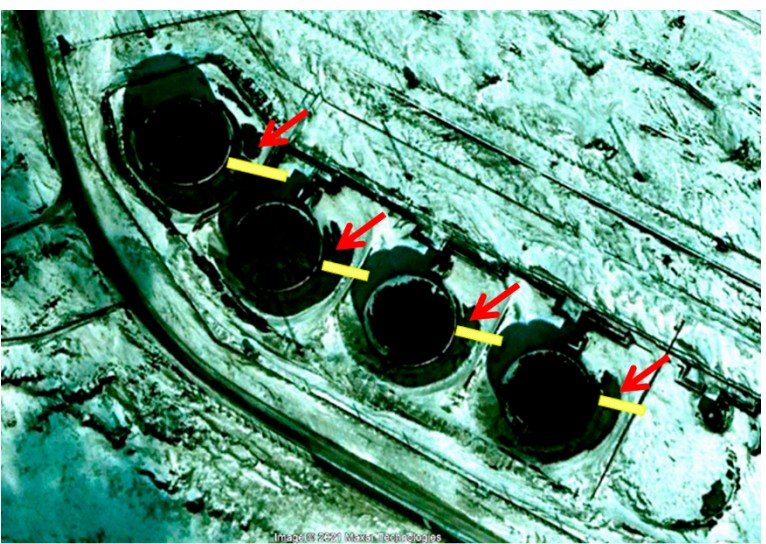

**Figure 13.** Google Earth image of oil tanks in early spring. Yellow strips mark the location of virtual horizontal plates of dihedrals (OB in Figure 3). Red arrows mark snow free patches near the tanks.

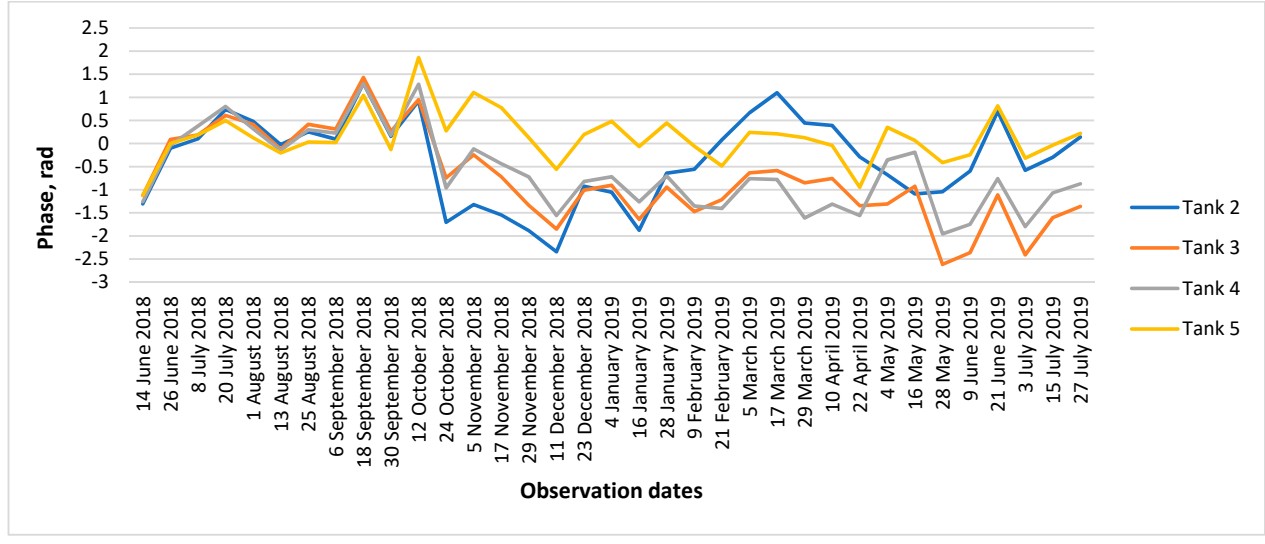

**Figure 14.** Accumulation of the phases for tanks 2–5 in 2018–2019.

Following the requirement of high temporal coherence, we presented here only three measurements for tanks 2 and 5 and eight measurements for tanks 3 and 4. The mean phase of the tank 2 signal is +0.06 rad, tank 3: +0.06 rad, tank 4: −0.17 rad, and tank 5: −0.49 rad. As can be seen, the phase values are within one rad tube, which is equivalent to the 0.4 cm potential mutual slant range displacement of the scattering surfaces. Though the mean phases obtained here are in contradiction with the phases from Figure 12 accumulated in July 2020, they may be considered more reliable because they are free from the corrupting influence of atmospheric precipitation. Additionally, the measurements of tank 5's phases do not confirm any centimeter-scale displacements to the tank's eastern wall and the surface

of the adjacent concrete base on the one-year time interval. At the same time, it is necessary to remember that a relatively low spatial resolution of radar data was used (~10 m) along with the multi-looking and filtering of the interferogram, which, of course, preclude the detection of the displacements in the areas with a lower spatial size, which might also be the reason for the damage to tank 5 in May 2020.

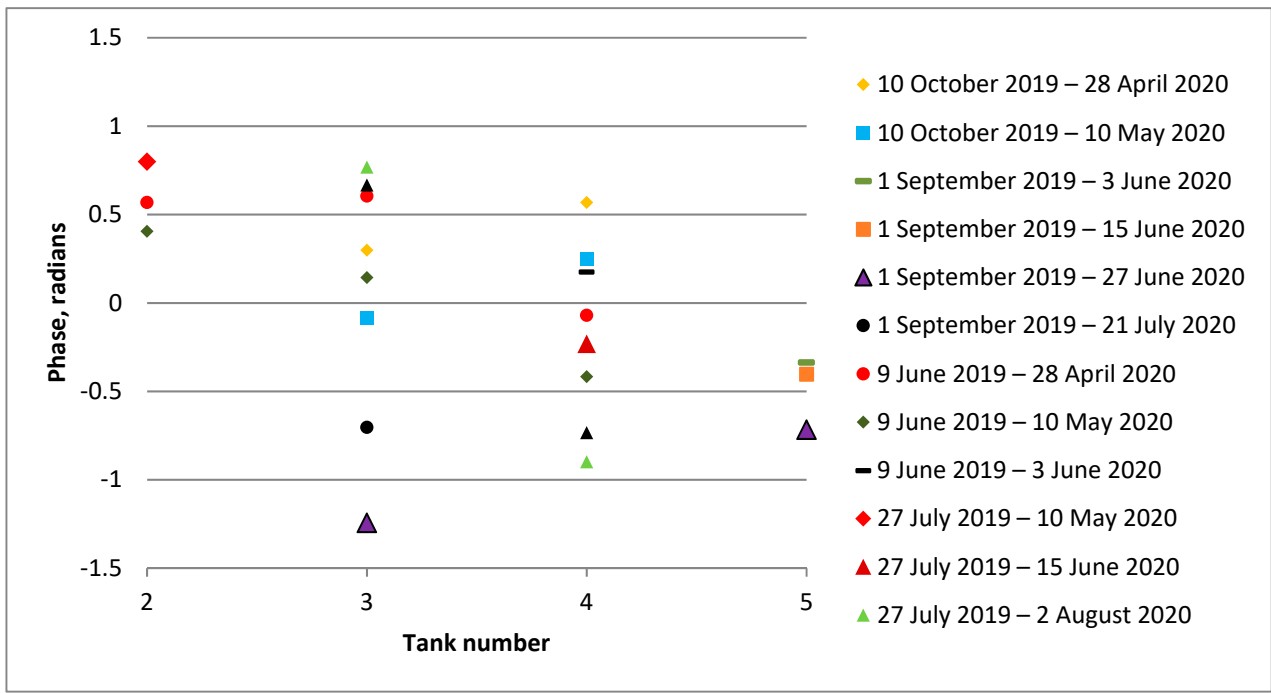

**Figure 15.** Phases of tanks 2–5 on the interferograms with long temporal baselines.

## 7. Conclusions

Monitoring the state of industrial infrastructure with the DInSAR technique in the far north is extremely difficult due to severe climatic conditions in the observation areas. Atmospheric precipitation (most particularly, snow), as well as freeze/thaw processes, corrupt the observations of the land cover dynamics in the DInSAR scheme of observations of the cylindrical objects such as fuel tanks that demonstrate the double-bounce mechanism of the backscatter. The models considered in this paper provide an explanation of two unusual phenomena observed—the abrupt 10–15 dB bounces of the tank's backscatter during wintertime and an absence of the monotonous interferometric phase growth that should be observed due to the snow accumulation (rise of snow water equivalent) near the tanks. As a result, an amplitude dispersion index of tanks 2–5 signals in the wintertime is too high from a formal point of view to consider the tanks as candidates for persistent scatterers and to apply the PS techniques for the monitoring of the tank's displacements.

Our experience with DInSAR observations of the fuel tanks at the Norilsk thermal power plant with Sentinlel-1 SAR during 2018–2020 shows that because of dominating the double bounce type of backscatter from fuel tanks it was necessary to provide special conditions for the observations. For example, to exclude the influence of snowfalls, the interferometric pairs should span the cold season and consist of the images acquired in the warm seasons of the year and preferably in dry weather conditions. The analysis of 13 interferograms spanning the wintertime period shows that the phases of all the tanks were within one rad tube during the 8–13 months interval, which is equivalent to the 0.4 cm potential mutual slant range displacement of the scattering surfaces. The troubleshooting phases of tank 5 do not confirm any centimeter-scale displacements of the tank's eastern wall, as well as the surface of the territory nearby, and do not support the hypothesis of permafrost thaw as the only reason for the accident.

The installation of corner reflectors on the roofs of the objects of interest may be the best solution for providing accurate DInSAR measurements. Taking into mind severe snowfalls and snow blizzards, the construction of the corner should prevent an accumulation of snow/water inside the corner. Additionally, one more (though offbeat) solution is to clear the snow within the narrow strip of the surface near the objects with the double-bounce type of backscatter before each SAR observation.

**Author Contributions:** Conceptualization, A.Z.; methodology, A.Z. and L.Z.; writing—original draft preparation A.Z. and L.Z.; writing—review and editing, A.Z. and L.Z.; data processing, L.Z.; project administration, A.Z.; funding acquisition, A.Z. All authors have read and agreed to the published version of the manuscript.

**Funding:** This research was carried out with the financial support of the Ministry of Science and Higher Education of the Russian Federation in the framework of the Agreement No. 075-01133-22-00.

**Institutional Review Board Statement:** Not applicable.

**Informed Consent Statement:** Not applicable.

**Data Availability Statement:** Not applicable.

**Acknowledgments:** The authors thank the European Space Agency for Sentinel-1 SAR data.

**Conflicts of Interest:** The authors declare no conflict of interest.

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
