# Peer review of "An Influence of Snow Covers on the Radar Interferometry Observations of Industrial Infrastructure: Norilsk Thermal Power Plant Case"

_remotesensing, doi:10.3390/rs15030654_

Round 1
Reviewer 1 Report
The manuscript presents the results of the study of snow covers influence on the interferometric measurements of the stability of industrial infrastructure in the vicinity of Norilsk city, Russia. Three models of snow layer within the dihedral proposed can help explain the deviations of signals amplitude and phase in the case of snowfalls occurred between radar observations, and used to minimize influence of snow on the interferometric measurements. However, the structure of the paper is rather chaotic, and here are some comments to improve paper further better:
1. Line 12-17 describes the object, data, and problems of the research. It is recommended that author briefly summarize the proposed method and reduce the text of the background.
2. Line 49-88 describes the application of differential SAR interferometry (DInSAR). However, the focus of the manuscript is to analyze an influence of snow covers on the radar interferometry observations of industrial infrastructure. Therefore, it is recommended to focus on the current research on DInSAR in radar interferometric observation of industrial infrastructure, and summarize the limitations and the innovation.
References:
[1] Samsonov, S. V., d'Oreye, N., González, P. J., Tiampo, K. F., Ertolahti, L., & Clague, J. J. (2014). Rapidly accelerating subsidence in the Greater Vancouver region from two decades of ERS-ENVISAT-RADARSAT-2 DInSAR measurements. Remote sensing of environment, 143, 180-191.
[2] Wang, L., Ma, H., Li, J., Gao, Y., Fan, L., Yang, Z., ... & Wang, C. (2022). An automated extraction of small-and middle-sized rice fields under complex terrain based on SAR time series: A case study of Chongqing. Computers and Electronics in Agriculture, 200, 107232.
3. Line 97, generally, Study area, Data and Methods are two independent contents, and it is recommended that author change it to "2 Study Area" and "3 Data and Method" or "2 Study Area and Data" and "3 Method".
4. Line 230-240 describes the interferometric processing of Sentinel-1 data using ENVI SARscape. I think it is data processing rather than method, so it should not be described too much here.
5. Line 245 proposed three snow layer models, and it is recommended that author combine this part with the previous interferometric method as the "Method" of the manuscript.
6. Line 315, generally, the title “Data processing and analysis of results” is rare. It is recommended that the author focus on summarizing the result of the paper, and change the title to “Result”.
7. “Discussion” is an important part of the paper, so it is recommended to add this part, and focusing on the uncertain factors and the innovation points in this research.
Author Response
Please find our replies in the file attached

Reviewer 2 Report
The reviewer would like to thank the authors for this thoughtful manuscript. This work has good potential. The authors are requested to put in some additional efforts to improve the quality of this manuscript.
Introduction
The authors have discussed a case of monitoring disaster for snow covered manmade infrastructure. Please extend this discussion to glacier related natural disasters as reported in the following article for highlighting the significance of the proposed methodology in forecasting such disasters. The authors are requested to elaborate more on the glacier-related disaster and cite the following article that reported a major disaster over the Himalayas, which largely impacted the local human livelihoods as well as manmade infrastructure.
-Shugar et al, A massive rock and ice avalanche caused the 2021 disaster at Chamoli, Indian Himalaya, Science, 2021.
SAR Based Snow Cover Identification
In order to precisely identify the tank locations it is important to accurately identify the snow cover. The authors are requested to discuss and cite the contributions of the following articles in the context of the application in this investigation.
-Muhuri et al., “Snow cover mapping using polarization fraction variation with temporal RADARSAT-2 C-band full-polarimetric SAR data over the Indian Himalayas”, IEEE JSTARS, 2018.
Please also include the possibility to observe snow cover using geostationary satellites.
-Qiao, H., et al., 2021. A New Geostationary Satellite-Based Snow Cover Recognition Method for FY-4A AGRI. IEEE JSTARS.
Orientation Angle Effect (wrt Satellite Look or Observation Direction)
The authors are requested to discuss the impact of orientation angle of the targets with respect to the satellite look direction. Please cite articles discussing the estimation of the orientation angle from the polarimetric data and compensation of the decomposed powers. Oriented structures are known to introduce pseudo volume power which can be misinterpreted as scattering from the snow volume in case of oriented structures which may not be covered with sufficiently thick layers of dry snow for causing volume scattering.
Figures
The authors are requested to declutter the figures. It is not necessary to provide every possible date for the duration of the investigation.
Conclusion
The authors are requested to list the key contributions in this section. At the moment the section is not detailed enough.
Author Response
please find our replies in the file attached

Reviewer 3 Report
The article deals with issues related to the monitoring of building structures using techniques based on image acquisition. The article still needs work, detailed comments below:
1. Abstract - a well-presented object of research as well as results and conclusions. However, there is a lack of an underlined purpose and methods used by the authors. From the abstract it is difficult to figure out which research methods the authors used and what is their proposal resulting from the research effect.
2. Section 1. Introduction - literature review does not lead to the identification of a research gap that the authors want to fill with their research. What is the novelty that the article presents?
3. Section 2. Study Area, Data and Methods Used - please explain all symbols used in formulas.
4. Section 2. Study Area, Data and Methods Used - "According to our estimations, the mutual tanks position is fixed within the 2-3 mm tube." – on what basis did the authors make the estimation? Did they build a displacement model? If so, what method? Such statements should be supported by appropriate explanations.
5. Section 2. Study Area, Data and Methods Used - in my opinion Fig. 3 is redundant.
6. Section 2. Study Area, Data and Methods Used - “Probable earlier displacements of the reservoir constructions could lead to an…” – on what basis do the Authors determine this probability? Were there signs of a possible design change? Was there monitoring at the facility? Such an assumption without research is wrong and affects further research.
7. Section 3.3. Heterogeneous snow layer within the dihedral - please check Eq. (12). In my opinion, this is incorrect.
8. Section 3.3. Heterogeneous snow layer within the dihedral - “Three above-mentioned models of the influence of the snow layer can help to…” – The authors do not explain why such snow cover models were accepted for research. The problem of model selection and previous research by other authors has not been presented in section 1. Introduction.
9. Section 4. Data processing and analysis of results - Fig. 8 - in my opinion, the graph should be constructed in a way that shows both variables: temperature and cover thickness. Currently, it is not known what one vertical axis represents. cm? oC?
10. Section 4. Data processing and analysis of results - Fig. 9 - dates on the horizontal axis should be corrected. The same remark applies to the other Figures. Fig. 15 is illegible.
11. Section 5. Conclusions - the part concerning the limitations of the use of technology is known (impact of climatic conditions and snowfall). There is no emphasis on novelty.
12. Cyrillic letters appear in the text. Please correct.
Author Response

(The authors gave the same response as above.)

Round 2
Reviewer 1 Report
1. Line 56-96 describes the application of differential SAR interferometry (DInSAR). However, this part is excessive, which affects the readability of the article, so it is recommended that the authors make deletions and focus on the research status of DInSAR in radar interference observation of industrial infrastructure.
2. Line 146-177 describes the characteristics of the radar cross section (RCS), but this is not part of the content of “2 Study Area”, and it is recommended to modify.
3. Line 262-264 shows Amplitude and Interferogram images, but lacks a legend.
4. Line 250, it is recommended that the authors change “Processing Methodology” to “Method”.
5. Line 407, it is recommended that the authors change “Processing results and analysis” to “Results and Analysis”.
Author Response
Answers to remarks of Reviewer
(Again, we appreciate the reviewer remarks that helped to bring our paper to a new level of quality. Our answers below are highlighted in bold)
- Remark: Line 56-96 describes the application of differential SAR interferometry (DInSAR). However, this part is excessive, which affects the readability of the article, so it is recommended that the authors make deletions and focus on the research status of DInSAR in radar interference observation of industrial infrastructure.
Answer: Yes, we changed the introduction section severely trying to focus on the application of DInSAR technique to the monitoring of industrial infrastructure, thank you.
- Remark: Line 146-177 describes the characteristics of the radar cross section (RCS), but this is not part of the content of “2 Study Area”, and it is recommended to modify.
Yes, OK, we presented this portion of the text as a separate section.
- 3. Remark: Line 262-264 shows Amplitude and Interferogram images, but lacks a legend.
Answer: We added the legend describing the link between intensity on the interferogram and phase values.
- Remark Line 250, it is recommended that the authors change “Processing Methodology” to “Method”.
Answer: Yes, we followed this recommendation and renamed the section to “Method”.
- Remark: Line 407, it is recommended that the authors change “Processing results and analysis” to “Results and Analysis”.
Answer: Yes, we followed this recommendation and renamed the section to “Results and Analysis”.
- We looked through entire text again and corrected some typing errors we found.
